# Hemicellulose and Nano/Microfibrils Improving the Pliability and Hydrophobic Properties of Cellulose Film by Interstitial Filling and Forming Micro/Nanostructure

**DOI:** 10.3390/polym14071297

**Published:** 2022-03-23

**Authors:** Yan Li, Mingzhu Yao, Chen Liang, Hui Zhao, Yang Liu, Yifeng Zong

**Affiliations:** 1College of Light Industry and Food Engineering, Guangxi University, Nanning 530004, China; 1916391015@st.gxu.edu.cn (Y.L.); yaomingzhu@st.gxu.edu.cn (M.Y.); liangchen@gxu.edu.cn (C.L.); zhh@gxu.edu.cn (H.Z.); 2016391055@st.gxu.edu.cn (Y.Z.); 2Guangxi Key Laboratory of Clean Pulp & Papermaking and Pollution Control, Guangxi University, Nanning 530004, China; 3State Key Laboratory of Biocatalysis and Enzyme Engineering, School of Life Sciences, Hubei University, Wuhan 430062, China; 4Guangxi Bossco Environmental Protection Technology Co., Ltd., Nanning 530000, China

**Keywords:** bagasse, nano/microfibrils, hydrophobicity, high pliability, high-consistency refiner

## Abstract

In this paper, nano/microfibrils were applied to enhance the mechanical and hydrophobic properties of the sugarcane bagasse fiber films. The successful preparation of nano/microfibrils was confirmed by scanning electron microscope (SEM), X-ray diffraction (XRD), fiber length analyzer (FLA), and ion chromatography (IC). The transparency, morphology, mechanical and hydrophobic properties of the cellulose films were evaluated. The results show that the nanoparticle was formed by the hemicellulose diffusing on the surface of the cellulose and agglomerating in the film-forming process at 40 °C. The elastic modulus of the cellulose film was as high as 4140.60 MPa, and the water contact angle was increased to 113°. The micro/nanostructures were formed due to hemicellulose adsorption on nano/microfilament surfaces. The hydrophobicity of the films was improved. The directional crystallization of nano/microfibrous molecules was found. Cellulose films with a high elastic modulus and high elasticity were obtained. It provides theoretical support for the preparation of high-performance cellulose film.

## 1. Introduction

Due to the extensive use of petroleum-based materials, a great deal of white pollution brings a critical menace to the ecotope [1]. In recent years, people have paid more and more attention to the development and application of green, sustainable, and renewable environment-friendly materials [2,3]. Cellulose films (i.e., nanofibrils film [4], microfibrils film [5], and nano/microfibrils film) had attracted intensive interest during the past decade due to its green sustainable nature and constant advances in micro and nanoscale patterning [6] for many applications of functional materials [7,8], including the filler [9], composite material manufacturing [10,11,12], packaging coating [13,14], medicine [15,16], high-performance green flexible electronics [17], and biotechnology [18].

Cellulose can be pretreated and machined to produce nanocrystalline cellulose (CNC), nanofibrils cellulose (NFC) [19], and microfibrils cellulose (MFC) [20]. It has unique attributes, such as enhanced capabilities [21], high mechanical strength, and adjustable self-assembly in an aqueous solution, due to its unique surface chemistry, size, shape, and high crystallinity [22]. At present, cellulose film research mainly focuses on the film properties of cellulose uniform system (i.e., uniform nanocellulose system, uniform microcellulose system), such that combining nanofibrils with renewable polymers could prepare partially degradable materials to improve material performance deficiencies and to address white pollution [23,24]. Shu et al. [25] prepared cellulose-based bioplastics, which were reorganized by the aggregation structure of cellulose. However, the hydrophobicity of regenerated cellulose membrane was not studied.

However, the application of cellulose films developed to date has been impeded by their poor mechanical performance and poor water resistance [26]. In the literature, a number of approaches have been reported to improve the increased hydrophobicity from cellulose films, such as low surface energy organic compounds [27], e.g., esterification [28], silanization [29], amidation [30], carboxymethylation [31], and fluoropolymer [32]. Unfortunately, this approach had a poor modification effect and resulted in problems related to organic pollution. The performance of nano/microfibril films dictates their hydrophobicity and pliability in relevant applications; however, utilizing the inherent structure of nano/microfibrils within the films for materials applications was an approach still in its infancy.

The method of the nano/microfibrils film for improving hydrophobicity and pliability, which aims to enhance the hydrophobic and pliability of the nano/microfibrils film without surface modification (i.e., unitary nano/microfibrils component), has not been reported. This study provides a simple pathway to improve hydrophobic and pliability properties of sugarcane bagasse nano/microfibrils film isolated by a high-consistency refiner and exhibiting a great potential for the further utilization of cellulose. We used enzymatic pretreatments to investigate the grinding of sugarcane bagasse fiber using a high-consistency refiner grinding to reproduce nano/microfibrils films to determine the mechanical and hydrophobicity properties. The main advantage compared to previous methods is its simplicity in preparation with less environmental pollution and a better economic benefit. The morphology, particle size distribution, structure, and transparency of the obtained nano/microfibrils were analyzed with a scanning electron microscope (SEM), an atomic force microscopy (AFM), a fiber length analyzer (FLA), a zetasizer nanoanalyzer, an X-ray diffraction (XRD), an ion-exchange chromatography (IC), and an ultraviolet and visible spectrophotometer (UV–Vis). The water resistance and mechanical properties of the nano/microfibrils films were analyzed by contact angle determination (CA), water vapor transmission rate (WVTR), and tensile tests.

## 2. Materials and Methods

### 2.1. Materials and Chemicals

Bleached sugarcane bagasse pulp was procured from Guitang Co., Ltd. (Guangxi, China). Celluclast (1.5 L, from *T. reesei*) was purchased from Novoxin Biotechnology Co., Ltd. (Beijing, China).

### 2.2. Methods

#### 2.2.1. Enzyme Pretreatment

Enzymatic hydrolysis pretreatment was carried out with untreated sugarcane bagasse samples. The fibers were treated with cellulase. The fibers and the appropriate amount of cellulase were mixed at a ratio of 1:500. A 1 mL enzyme solution was added to every 500 mL 30 wt% bagasse samples. The enzyme activity was 16 FPU/mL. The most suitable pH was 6.0–8.0. After constant temperature stirring at 50 °C for 30 min, the enzyme activity was eliminated with a constant temperature water bath at 100 °C for 10 min and washed with distilled water multiple times to remove residual enzyme liquid.

#### 2.2.2. Preparation of Nano/Microfibrils by Mechanical Grinding

The enzyme pretreated sugarcane bagasse pulp was diluted to 30% (*w*/*w*) with distilled water and ground with a high-consistency refiner at a disc gap of 0.1 mm, and with 10, 20, 25, 30 rounds of grinding. After grinding, a mixture of nanocellulose and micro-cellulose was stored in a refrigerator at 4 °C for subsequent analysis.

#### 2.2.3. Preparation of Nano/Microfibrils Film

A nano/microfibrils film was formed by pouring nano/microfibrils into a polystyrene template by the flow-edge method, which could be prepared by drying nano/microfibrils in oven at 40 °C for 24 h, as shown in Figure 1.

### 2.3. Characterization

To evaluate the properties of samples and understand the changes caused by mechanical processing, a raw material analysis of fiber components was performed on the specimens. Then, the hydrophobic and mechanical properties of cellulose film samples were analyzed.

#### 2.3.1. Characterization of Morphological Features

AFM (SI-DFP2, Hitachi, Tokyo, Japan) and SEM (SU8220, Hitachi, Tokyo, Japan) are often combined with cellulose image analysis. In the SEM test, samples were observed at a voltage of 10 kV after spraying gold under vacuum for 90 s. The magnification was 500× and 2000×, respectively. AFM could evaluate their surfaces and morphological changes at the test pressure of 299 kHz, an accelerating voltage of 8.1 V, and a cantilever elastic coefficient (C) of 32 N/m.

#### 2.3.2. XRD Analysis

The nano/microfibril films were cut into small pieces of 1.5 cm × 1.5 cm and used XRD (Miniflex600, Rigaku, Tokyo, Japan) under Cu_Kα_ ray radiation (λ = 0.15418 nm) for inspection. The scanning range was 2θ = 5°–35°, and the speed was 5°/min. The tube pressure level and the tube flow rate were 40 kV and 30 mA, respectively. Segal’s empirical formula (*CrI*) was used to calculate the crystallization index of the samples [33].
*CrI* (%) = (*I*_002_ − *I*_am_)/*I*_002_(1)
where, *CrI* is the crystal index, *I*_002_ is the diffraction intensity level obtained when 2θ = 22.6°, that is, the diffraction intensity of the crystal region. *I*_am_ is the diffraction intensity level obtained when 2θ = 16.0°, that is, the diffraction intensity of the amorphous region.

#### 2.3.3. Particles Size Analysis

The particle size and its distribution of the nano/microfibril suspension were measured by Malvern Zetasizer nano (ZS90X, Malvern Panalytical, London, UK). A 0.5 wt % nano/microfibril suspension was tested after magnetic stirring for 12 h.

The raw cellulose properties including length-weighted distribution (Ln), weight-weighted distribution (Lw), and fines were measured with a fiber length analyzer (FLA, Kajaani FS-300, Metso Automation, Helsinki, Finland).

#### 2.3.4. IC Analysis

The content of hemicellulose sugar in sugarcane bagasse pulp was determined by ion chromatography (ICS-5000+SP, Thermo Science, Sunnyvale, CA, USA). The chromatographic column was a PA20, 3 × 150 mm, and the protection column was a PA10, 4 × 50 mm. A mix of 80% Milli-Q water and 20% NaOH was used for the mobile phase. The flow rate was 0.3 mL/min. Specific steps conformed to the National Renewable Energy Laboratory standard method TP-510-42618.

#### 2.3.5. UV–Vis Analysis

UV–Vis (SPECORD-PLUS-50, Analytik Jena, Berlin, Germany) was used to measure the transmittance of the samples. The film sample was carefully cut into a rectangle of 40 mm × 9 mm and placed in a quartz cuvette 25 cm from the entrance of the integrating sphere. A quartz cuvette was placed as a blank reference. The wavelength range of the sample was 190–1100 nm, and the transmittance of the visible band (400–800 nm) was analyzed.

#### 2.3.6. Water Resistance Analysis

The water contact angle of the sample was measured using a Drop Shape Analyzer (DSA100, KRUSS, Berlin, Germany) to evaluate the surface hydrophobicity of the film samples. The film samples were fixed on the glass slide with double-sided tape. An automatic pipette was used to carefully apply a drop (4 μL) of distilled water onto the film surface. Parallel tests were performed six times and the results were averaged.

The WVTR (W3/031, Labthink, Jinan, China) measurements were performed according to the standard ASTM Standard E96/E96M-05 for cup method water vapor permeability [34] testing at 25 °C with 90% RH. Weights were monitored every 30 min until constant.

#### 2.3.7. Mechanical Properties

The mechanical properties consisting of folding endurance, elasticity modulus, and elongation at break of the films were measured with a universal testing system (3367, INSTRON, Sunnyvale, CA, USA). The test was conducted at a crosshead speed of 2 mm/min at 23 °C and 55% relative humidity. The sample length was 50 mm and the width was 10 mm.

## 3. Results

### 3.1. SEM Analysis of Fibers after Different Stages of Treatments

The SEM analysis of the fiber suspension obtained after different grinding stages are shown in Figure 2a–e. The obtained images exhibit substantial differences in surface and morphological changes. In C-0 without grinding treatment, the fiber bundle structure was compact, and the fibers were relatively compact, such as in Figure 2a. The images obviously show that the sugarcane-bagasse-bleached pulp fibers were nearly smashed into nano/microfibrils after the high-consistency refiner, although a few microfibrils bundles with correspondingly larger fibers still exist. After the mechanical processes, the tightly bound fibers divided into smaller fiber bundles, as shown in Figure 2b–e. In the higher right corner of Figure 2b–e are 2 μm images, which show that a large number of nano/microfibers were partly stripped from the surface due to the effects of grinding. The fiber diameters decreased with each treatment grinding times, and high-speed grinding gradually declined the fiber length and layers’ thickness. After grinding more than 25 times, we observed that cellulose fibers were almost completely decomposed into nano/microfibrils aggregates around the fiber (Figure 2d). It formed many fine cellulose aggregates in Figure 2c,d, possibly due to the soluble hemicellulose being exposed during the grinding process and because hemicellulose was loosely bound to the fiber surface. Echoing the results of Tenhunen [35], xylose, one of the principal components of hemicellulose from sugarcane bagasse, had an affinity for cellulose and readsorbs to the cellulose surface during the grinding process. The presence of xylan with a negative charge facilitates the liberation of fibrils from the pulp by generating repulsion between fibrils. Accordingly, in the case of higher levels of hemicellulose in sugarcane bagasse, the preparation of nano/microfibers by the high-consistency refiner became more efficient.

### 3.2. Particle Size and Components Analysis

The particle size and particle size distribution of the nano/microfibrils and untreated fibers (C-0) suspension were measured by a zetasizer nano analyzer (Figure 3) and FLA (Table 1). Under the action of mechanical force, bagasse fiber was easy to crack along the axial direction and formed fibrillary cellulose. Fibrillated ultrafine fiber was an anon-uniform system composed of fibers of different sizes and shapes. Without homogenization, the fines fiber (0–0.2 mm) content was approximately 46.16% and when grinding 10 more times, the C-30 fines fiber content was approximately 53.52% (Table 1). L(n) and L(w) of untreated fibers were 0.34 mm and 1.20 mm, respectively; after 10 grindings, L(n) was 0.29 mm and L(w) was 0.86 mm. These data indicate that grinding had an important effect on the L(w) of fibers, especially in the longitudinal direction.

Fibrillated ultrafine fiber is a complex system, which contains a large number of submicron and nanofibers in the range of ultrafine fibers. The zetasizer nano analyzer was used to analyze the particle size range of nano/microfibrils in the fibril suspension system (Figure 3). Different sizes and size distributions of fibrillated ultrafine fiber had an important influence on the film-forming properties. After grinding 10 times, the particle size distribution was in the range of 539–1591 nm. The particle size distribution of C-20 and C-25 fibers had three distinct peaks, one at 100–500 nm, another at 500–1900 nm, and the last one at 3500–6500 nm (Figure 3). This indicates that the cellulose suspension was a multisize mixed system. After grinding 30 times, the particle size distribution was mainly concentrated in the range of 130–400 nm, and the fiber suspension system was relatively uniform. Therefore, compared with unground bagasse cellulose, the size distribution of ground bagasse cellulose was more extensive and richer. The average length of unmilled cellulose was about 650–2170 μm. After milling, the cellulose system had a lot of nanocellulose (1–10 μm) and microcellulose (10–200 μm), and some millimeter-grade cellulose (200–7000 μm). There were many sizes of cellulose, such as nanometer (13–36%), micron (17–32%), and millimeter (46–54%), in the cellulose suspensions. The fiber size distribution of the fibrillated ultrafine fiber was more abundant after 25 times of grinding. The fibrillated ultrafine fiber suspension system of C-30 was relatively uniform. It was feasible and highly efficient to prepare a hybrid system of fibrillated nano/microfibrils with the high-consistency refiner.

The analysis of the sugar composition content of fibers was obtained by ion chromatography. Since the hemicellulose in the bagasse bleaching pulp was mainly xylose, there was also a small amount of arabinose and galactose. As shown in Table 2, all samples contained three carbohydrates: glucose, xylose, and arabinose. Compared with C-0 and C-E, the glucose and arabinose contents were reduced after grinding; instead, the xylose content was increased. The hemicellulose contents had a significant effect on the film formation, wettability, and mechanical properties of the film, through the interaction between hemicellulose and cellulose [36]. The hemicellulose content of C-0 and C-E was 21.21%, and 20.97%, respectively, which indicated that cellulase treatment did not affect the dissolution of hemicellulose. The content of xylose and arabinose in the cellulose suspension after grinding decreased slightly, which proved that the hemicellulose between the secondary wall fiber bundles was separated in the nano/microfibrils suspension after grinding, and a part of the hemicellulose polysaccharide was soluble in water, so the measured xylose and arabinose content decreased. The results prove that exposing sugarcane bagasse to suitable grinding times when grinding with high-consistency refiner plays a key role that can increase the hemicellulose content in microfibers.

### 3.3. Crystallinity of Fibers after Different Treatment Stages

Figure 4 shows crystallinity of nano/microfibrils obtained after various treatments. Two evident diffraction peaks were obtained at 16.0°and 22.5°, corresponding to the (110) and (002) crystal planes of a prototypical cellulose I structure [37]. The highest crystallinity of the cellulose (C-0) was 53.4%. After cellulase treatment of the raw materials (C-E), the crystallinity was 55.2%. There was no significant change in crystallinity after cellulase treatment, indicating that cellulase pretreatment had little effect on the crystallinity of cellulose. Crystallinity indices of C-10, C-20, C-25, and C-30 were 45.0%, 47.3%, 48.2%, and 48.3%, respectively, increasing a little with each grinding time. This may be related to the different content of hemicellulose in cellulose suspension. Hemicellulose was an amorphous polymer. Therefore, the XRD analysis uncovered that fibers had a striking decrease in crystallinity index after the high-consistency refiner pretreatment. The XRD analysis indicated that the crystallization peak of fibers did not change through the mechanical process stages. Part of the hemicellulose was exposed to the fiber suspension during the high-concentration refining process, thereby reducing the crystallinity. However, the number of grindings had little effect on crystallinity.

### 3.4. Optical Transparency of Nano/Microfibrils Films

Transparency was an important advantage in the use of packaging materials. In Figure 5, the optical properties of the nano/microfibrils films were surveyed and compared with photos from a camera. Moreover, the impact of grinding times on the transmittance properties of nano/microfibrils were researched using a UV–Vis method in Figure 5a. At a 400–800 nm wavelength, the light transmittance of C-0 without mechanical treatment was about 0.97% at room temperature. Compared with C-0, all nano/microfibrils films exhibited high levels of transparency and it significantly increased the transmittance to a maximum of 18.58% (C-25) after mechanical grinding by increasing the nano/microfibrils content. This proved that the fine fibers play a decisive role in the transparency of the film. The higher the fine fiber content, the higher the transparency. However, the transparency of the C-30 film was reduced a little, by 2.36%, as the hemicellulose could enhance the reflected light, and decrease the transmittance of the film [38].

SEM images of the film cross-section showed that the surface of the untreated fiber film was nonuniform with a large number of fiber fragments and a big porosity between fibers (Figure 5b). However, a more uniform surface morphology, smaller fibers and nano/microfibrils bundles, and a more compact structure of the cross-section were observed on the films after grinding (Figure 5c). This phenomenon indicated that as the grinding time increased, more nano/microfibrils were obtained, and denser and better structure films were formed. The tight overlap between the fibers eliminated the scattering and reflection of light inside the film, so that the transmittance of the film was significantly increased. Therefore, the densification and refinement of the nano/microfibrils structure directly led to the higher transmittance and strength of films [39].

### 3.5. Hydrophobic Properties and Principle of Nano/Microfibrils Film

The surface hydrophobicity of raw materials and nano/microfibrils films were evaluated through the measurements of the surface contact angles, and the hydrophobic mechanism of the cellulose membrane was analyzed (in Figure 6). The CA of the C-10 film was 98.7°, indicating that the nano/microfibrils film was hydrophobic (CA > 90°). With the increase of grinding times, the hydrophobicity first increased and then decreased (in Figure 6a). The CA of the nano/microfibrils film was the highest at 113° after grinding 25 times (Figure 6a). This may be due to the cross-linking of hemicellulose and nano/microfibrils during the film-forming process. Hemicellulose adsorbed on the pores of nano/microfibrils as fillers, and the surplus hemicellulose on the surface of the cellulose to form a large number of micro/nanostructures. Thus, it had high hydrophobicity. The surface wettability of the material depended on the surface chemistry and the surface micro/nanostructure [40]. These two parameters determined the level of adhesion between the droplet and the surface. The nano/microfibrils film was hydrophobic after grinding (in Figure 6b). The primary reason was that part of the hemicellulose existed in the form of single molecules or colloidal aggregates in the cellulose suspension after the grinding treatment. Then, hemicellulose diffused and adsorbed to the cellulose surface to form nanoparticles during the film-forming process at 40 °C. The film surface owned a certain roughness and was hydrophobic.

In Figure 6c, many dense white micro/nanostructures on the surface of the film exist, which may be because the exposed hemicellulose was adsorbed on the surface of nanocellulose driven by entropy [41]. As previously [42] reported, the xylan is likely to assemble into nanoparticulate aggregates in solution first and it was deposited like this onto the cellulose fibrils, forming a large number of white micro/nanoprotrusions consistent with the C-25 film surface 3D image analysis in Figure 6d,e. The surface roughness (Sa) was 4.493 nm.

However, the CA measurements of C-30 film was reduced to 95°, which could be due to the increase of fine fiber, which broke the balance between hemicellulose and fine cellulose in the fiber suspension system. After hemicellulose adsorbed on the pores of nano/microfibrils, the surplus hemicellulose on the surface of cellulose decreased. Because the micro/nanostructure formed on the surface of the film decreased, the contact angle of C-30 slightly decreased. This phenomenon indicated that the particle size of the fiber reduced after grinding, and that the nano/microfibrils became the principal component of the fiber suspension. This showed that the size of fibrils determined the macrostructure properties of the film and the hemicellulose content in the suspension determined the microstructure properties of the film. The micro/nanostructure formed by the exposure of the hemicellulose played the leading role in affecting the cellulose film hydrophobicity properties.

### 3.6. Barrier Performance of Nano/Microfibrils Film

The water vapor barrier performance of the substrate side (contacting the polystyrene template) as a moisture barrier was studied by the water vapor transmission rate (Figure 7a). The water vapor transmission rate of the untreated cellulose film was about 2426.59 g/(m^2^·24 h). It significantly improved the barrier properties of the film after grinding treatment. The fine fiber content of the C-25 film was 50.96%, its permeability pass rate was 1038.16 g/(m^2^·24 h), and the barrier performance was about twice as high as before. The water vapor transmittance was not a linear function of the fine fiber content, although, with the increase of the fine fiber content, the water vapor transmittance decreased (Figure 7b). This highlighted that the content of fine fibers increased the barrier property of the film. The water vapor transmission rate of the C-25 film was as high as 1038.16 g/(m^2^·24 h) under these conditions. According to previous studies, biodegradable films have been studied to use nanocellulose as a coating. Compared with the moisture resistance of nanocellulose film, the WVTR of CNF-coated paper was about 300 g/(m^2^·24 h), and the moisture resistance was improved by 55%. The water vapor permeability of CNF was about 960–980 g/(m^2^·24 h) [43]. It was the same as that of traditional CNF film. However, the preparation process of nano/microfibrils film in this study was simple. After grinding treatment, the barrier properties of the film were significantly improved. This phenomenon was primarily due to the fact that the fine fibers became the main component of the fiber slurry after the grinding treatment. During the film-forming process, microfibers and nanofibers were inserted between the large fiber bundles, the pores between the large fibers were filled with fine fibrils, which separated a part of the hemicellulose into the nano/microfiber suspension during the grinding process. There was adsorption between hemicellulose and cellulose fibers, and hemicellulose was used as the adhesive to fill the pores between the fine fibers again in the film-forming process, so the barrier property of the film was improved remarkably. However, when the content of fine fibers was 53.52%, the water vapor transmission rate of the film increased. This was because the increase of fine fibers was much greater than the increase in hemicellulose, and a large number of pores between the fine fibers could not be filled by the increased hemicellulose, resulting in a slight decrease in the barrier properties of the film. Therefore, fine fiber and hemicellulose were the key factors to improve the water vapor barrier performance of nano/microfiber films.

### 3.7. Mechanical Properties of Films

The thickness, folding endurance, and elastic modulus of the nano/microfibril films are recorded in Figure 8. C-10 displayed a higher thickness, lower folding endurance, and lower elastic modulus, which were 31.1 μm, 1.59, and 2671.18 MPa, respectively. Compared to the films after different treatment stages, C-30 displayed a lower thickness, higher folding endurance, and higher elastic modulus, at 25.9 μm, 1.50, and 4140.60 MPa, respectively, which indicated that the film was more compact and flexible. In the lower right corner of Figure 8 is a paper airplane made of C-30 film, which showed high pliability. This was because the content of nanocellulose in the cellulose system was 36.34% after grinding 30 times, which was about triple the content of nanocellulose in the 10-time grinding system. The elastic modulus of the films increased by about 62% when the content of nanocellulose was 13–36%. The tensile strength tripled. Moreover, because the cellulose particle size distribution was relatively rich, nanocellulose and microcellulose [(0–9)] could provide a certain degree of strength and stiffness for the film. The folding resistance decreased from 1.7 to 1.5 and decreased with the decrease of large-size cellulose. This showed that a larger size of cellulose could make the film have a certain degree of flexibility. Therefore, the thin film prepared by this method had not only a certain strength and rigidity but also certain folding properties.

Then, according to Figure 8, the C-20, C-25, and C-30 films had a stress platform at about 0.5–1.5% strain, with yielding remaining unchanged with the increase of strain during the stretching process, which was a nonuniform stretching stage. The elastic moduli were 3866.62 MPa, 3760.51 MPa, and 4140.60 MPa, respectively. This was because the film contained fibers of different sizes. During the stretching process, the crystal structure inside the film changed (Figure 9). Under the action of an external force, the crystal lattice molecules of micron-sized fibers and nano-sized fibers began to adjust and orient in the direction of the external force, destroying large-sized cellulose [44,45]. The reduced tensile strength of the lattice failure offset the increased tensile strength after the orientation of the nano/microfibers molecules, forming a curved platform area [46]. The findings indicated that the increase of a certain number of fine fibers could not only increase the hardness of the film, but also could give the film flexibility and plasticity. After passing through the plateau area, the C-25 and C-30 films appeared in a strain hardening stage, the stress rising sharply. This was because the molecules oriented under the stronger stress were highly oriented, and the materials formed a new higher-order crystal structure [46]. The tensile strength of C-25 and C-30 could be as high as 31.21 MPa and 32.70 MPa.

## 4. Conclusions

In this study, a novel process was proposed to obtain a hydrophobic sugarcane bagasse film with high pliability. The C-25 film had the best comprehensive performance, with better hydrophobic properties (CA = 113°) and a better elastic module of 4140.60 MPa. Fibrils with different hemicellulose and nano/microfibrils content were prepared by adjusting the number of times of the grinding. In addition, unique hydrophobic properties of films were obtained, due to the hemicellulose adsorption in the temperature-driven film-forming process. A better hydrophobic effect and high pliability were attained, and the adsorption effect of hemicellulose was effectively demonstrated. This research greatly improves the practical value of nonuniform cellulose in the field of packaging materials, which lays a foundation for the preparation of functional biomaterials without modification.

## Figures and Tables

**Figure 1 polymers-14-01297-f001:**
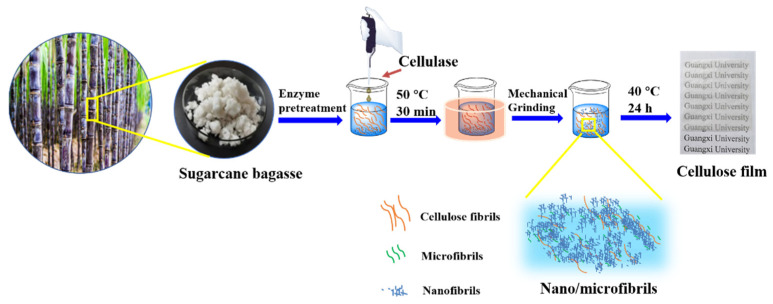
Schematic representation of process for preparing nano/microfibril film.

**Figure 2 polymers-14-01297-f002:**
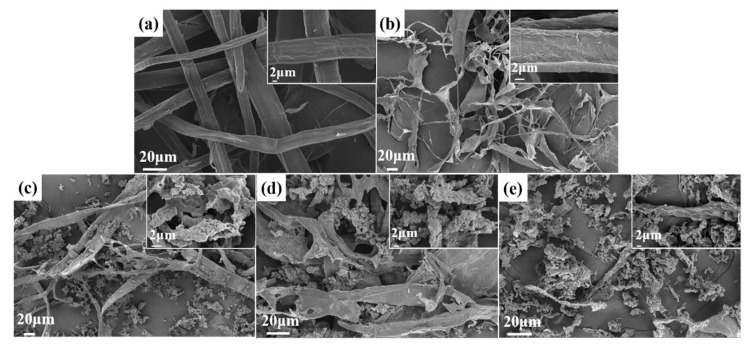
SEM micrographs of (**a**) the raw material C-0, (**b**) 10 times ground C-10, (**c**) 20 times ground C-20, (**d**) 25 times ground C-25, and (**e**) 30 times ground C-30. C-0 represents raw pulp fibers and C-10, C-20, C-25, and C-30 represent fibers that have been grounded 10, 20, 25, and 30 times by a high-consistency refiner, respectively.

**Figure 3 polymers-14-01297-f003:**
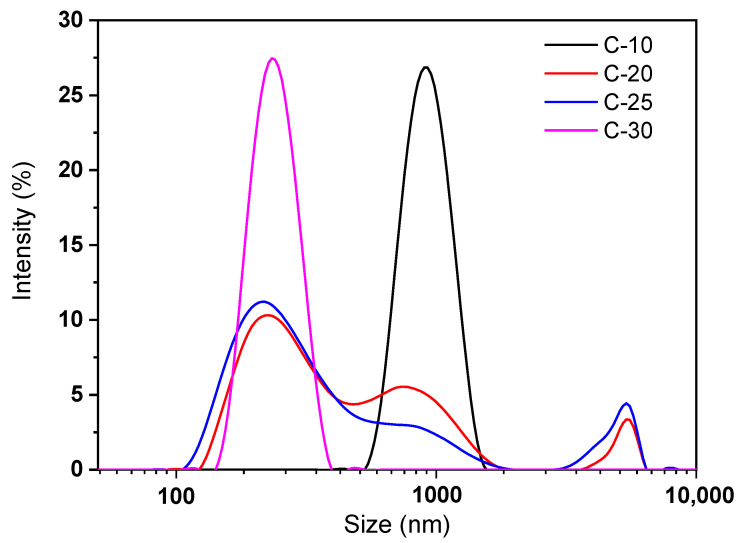
The particle size distribution of fibers: fibers ground 10, 20, 25, and 30 times fibers with a high-consistency refiner, respectively.

**Figure 4 polymers-14-01297-f004:**
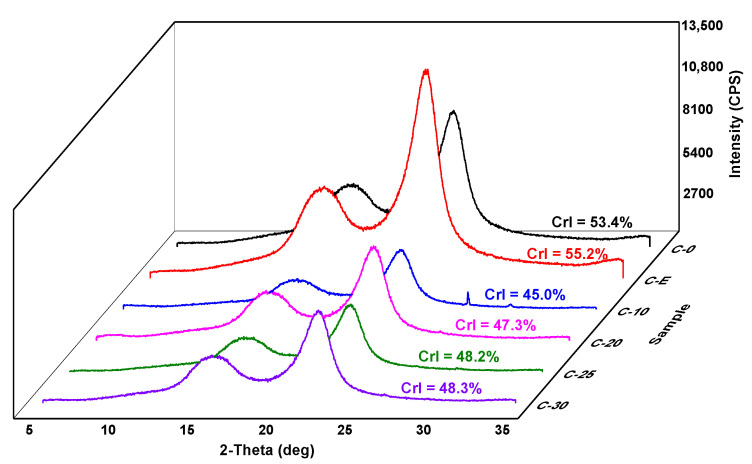
XRD patterns of nano/microfibrils films with different treatment stages.

**Figure 5 polymers-14-01297-f005:**
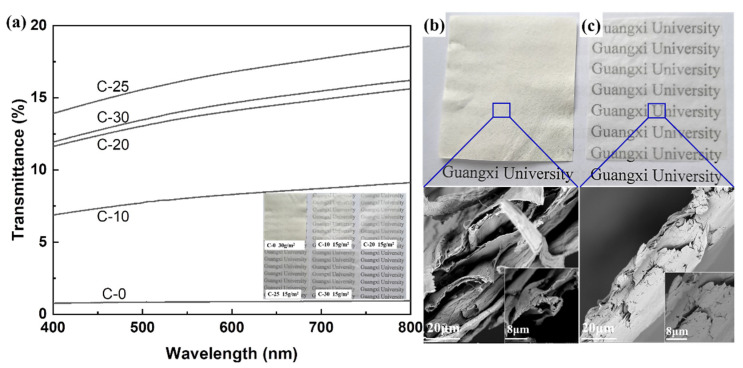
(**a**) Photo and optical transparency of films after different treatment stages, and (**b**) SEM micrographs and digital photographs of the raw material of C-0 film and (**c**) C-30 film after mechanical treatment.

**Figure 6 polymers-14-01297-f006:**
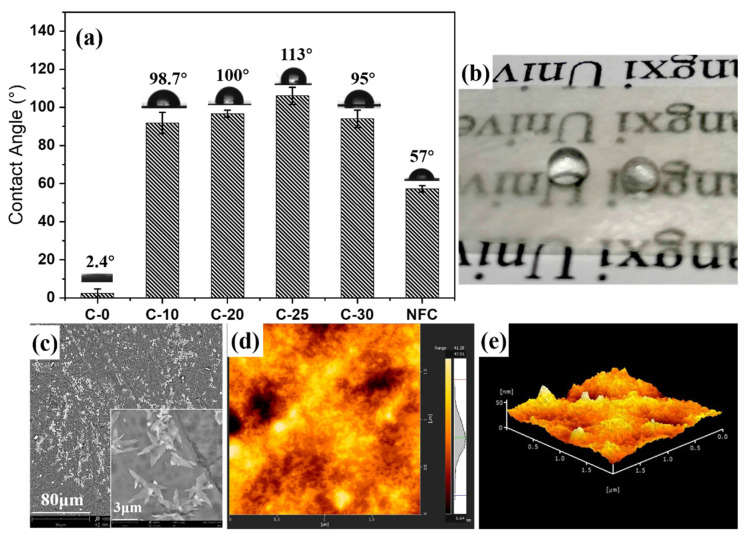
(**a**) Contact angle of nano/microfibrils films, (**b**) digital photographs of C-25 film hydrophobicity, (**c**) SEM micrographs of the C-25 film, (**d**) AFM surface topography (2 × 2 μm^2^), and (**e**) 3D image of the C-25 film.

**Figure 7 polymers-14-01297-f007:**
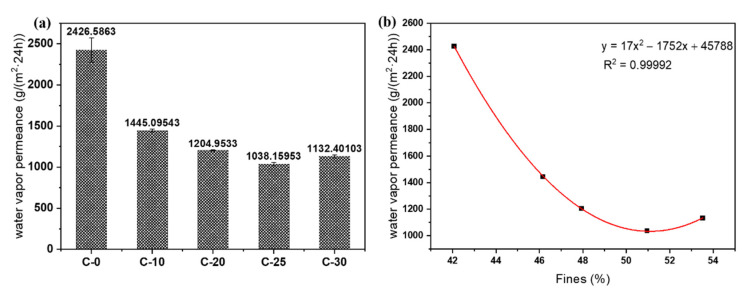
(**a**) Water vapor transmittance of nano/microfibrils films. (**b**) Fines (x) as a function of water vapor transmission rate (y). The solid line represents the nonlinear fit, with correlation coefficient R^2^  =  0.99992.

**Figure 8 polymers-14-01297-f008:**
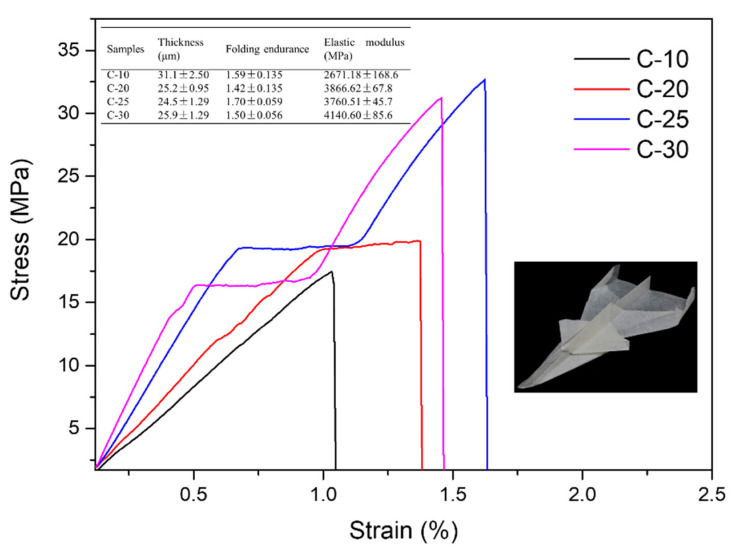
Stress–strain curves of the nano/microfibrils-based films.

**Figure 9 polymers-14-01297-f009:**
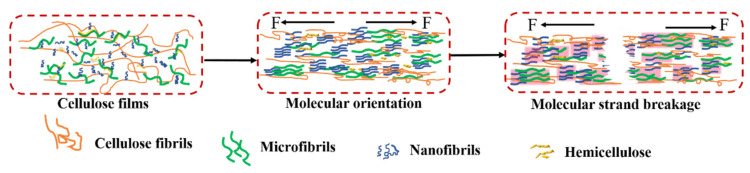
Schematic of tensile mechanism of films.

**Table 1 polymers-14-01297-t001:** Fiber size of different treatment stages.

	C-0	C-10	C-20	C-25	C-30
L(n), mm	0.34 ± 0.08	0.29 ± 0.02	0.28 ± 0.07	0.27 ± 0.02	0.27 ± 0.03
L(w), mm	1.20 ± 0.07	0.86 ± 0.04	0.80 ± 0.03	0.80 ± 0.05	0.78 ± 0.08
Fines, %	42.08 ± 1.46	46.16 ± 1.21	47.94 ± 0.98	50.96 ± 1.82	53.52 ± 1.63

**Table 2 polymers-14-01297-t002:** Sugar composition of fibers after different treatment stages.

Sample	Neutral Sugars and Acidic Oligomers (%)		
	Glucose	Xylose	Arabinose
C-0	74.95 ± 4.43	21.21 ± 0.90	1.16 ± 0.0006
C-E	75.48 ± 5.48	20.97 ± 0.56	1.14 ± 0.0006
C-10	72.64 ± 5.32	19.52 ± 1.52	1.05 ± 0.0006
C-20	72.07 ± 8.89	17.46 ± 1.45	1.01 ± 0.0011
C-25	73.02 ± 7.65	17.16 ± 0.96	0.86 ± 0.0018
C-30	73.39 ± 4.09	18.66 ± 0.98	0.96 ± 0.0004

## Data Availability

The data presented in this study are available in the manuscript’s figure.

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
