# Peer review of "Hemicellulose and Nano/Microfibrils Improving the Pliability and Hydrophobic Properties of Cellulose Film by Interstitial Filling and Forming Micro/Nanostructure"

_polymers, 2022, doi:10.3390/polym14071297_

Round 1

Reviewer 1 Report

In this work by Yao and co-workers, the authors prepared cellulose films and characterized their properties. The submission is relevant as it nicely matches the aims and scope of Polymers. However, some issues should be addressed before the paper can be accepted for publication. Please find the suggestions given below:
1) "The film sample was carefully cut into a rectangle of 40 mm ×9 mm and placed in a quartz cuvette 25 cm from the entrance of the integrating sphere. Taking air as a blank reference. The wavelength range of visible light is 380-800 nm." - in this case, it would be more appropriate to have an empty cuvette as a reference, would not it?
2) SEM micrographs - insets are barely visible, images are given at different magnification, there are no professional scale bar markers. Fig. 6c is barely visible.
3) There is only one stress-strain curve for each sample type in Fig. 8. It would be good to provide in SI at least three curves for each specimen to be able to verify how repeatable these measurements are. 
4) It would be useful to provide TGA results for the produced samples.

Reviewer 2 Report

The manuscript entitled "Hemicellulose and nano/micro-fibrils improving the pliability and hydrophobic properties of cellulose film by interstitial filling and forming micro/nano-structure" is well written and should be interesting for the nanocellulose community. The manuscript can be accepted after minor revision based on the following concerns-

  1. the authors describe the effect of crystalline domains orientation based on mechanical stress/strain ( line: 369- 385). The schematic representation is also logical; however, scientific evidence is absent here. The authors can include 2D XRD data or they can include an optical image with linear polarized microscopy. If they do not have any facilities for both of them, they must include references to validate their assumption.

  1. Following reference should include in the introduction section, along with references 9-11
  • Cellulose nanocrystal manufacturing process: Chowdhury, R. A., Clarkson, C., & Youngblood, J. (2018). Continuous roll-to-roll fabrication of transparent cellulose nanocrystal (CNC) coatings with controlled anisotropy. Cellulose, 25(3), 1769-1781.
  • Packaging coating: Chowdhury, R. A., Nuruddin, M., Clarkson, C., Montes, F., Howarter, J., & Youngblood, J. P. (2018). Cellulose nanocrystal (CNC) coatings with controlled anisotropy as high-performance gas barrier films. ACS applied materials & interfaces, 11(1), 1376-1383.

Reviewer 3 Report

The refrences are not adjusted to MDPI rules.

In the introduction part of the manuscript I am missing the recent devlopments in this area.

Line 73: should T. reesei be written in  italic?

Lines 7783: Do you have any information about the enzyme activity?

Despite the fact that some analysis used are well known (AFM, SEM, IC) you should provide full name (abbreviation in bracket) when mentioned for the first time. Afterwords you can use only abbreviation.

Lines 120-125: what about used column? any specific?

Line 130-131: The wavelength range of visible light is 380-800 nm. - Yes it is true but the measurement were performed in the UV -VIS spectral range from 380-800 nm.

Line 136: how many water droplets were taken? Only one is not enough!

Conclusion section should be a little bit more expanded with some new findings related to this study.

Round 2

Reviewer 1 Report

Thank you but the provided response is not satisfactory. 

1) "The film sample was carefully cut into a rectangle of 40 mm ×9 mm and placed in a quartz cuvette 25 cm from the entrance of the integrating sphere. Taking air as a blank reference. The wavelength range of visible light is 380-800 nm." - in this case, it would be more appropriate to have an empty cuvette as a reference, would not it?
R: Yes you are right. Thanks very much for your comments, we have made the following changes: A quartz cuvette was placed as a blank reference. The wavelength range of visible light is 380-800 nm.

Please do more research and present how the change of reference influenced the shape of the spectra. Simply changing the description is just a way to cover mistakes.

2) SEM micrographs - insets are barely visible, images are given at different magnification, there are no professional scale bar markers. Fig. 6c is barely visible.
R: Thanks very much for your comments. We have modified the size and clarity of SEM images in the manuscript and added rulers to the SEM images.

Firstly, ribbons with unnecessary and invisible information should be removed. Secondly, the authors provide wrong scale bars in:

  • Fig. 2a (inset)
  • Fig. 2b (both panels)
  • Fig. 2c (main panel)
  • Fig. 2d (inset)
  • Fig. 2e (inset) 

I ask the authors again to provide images recorded at the same magnification with correct scale bars.

Reviewer 3 Report

Please write the Si units as standard - mililitres should be written as mL not ml. This should be corrected in the whole manuscript.

Regarding the anwers you have given, some things in the answers to reviewer must be added to your manuscript, specificly related to the used column in IC analysis and explanation of UV - Vis analysis. You have section UV-Vis analysis, but in the manuscript you mentioned only the visible range. This is misleading information and as written in this way it is wrong. Thus, the answers you have written, should be added in the manuscript.

Round 3

Reviewer 1 Report

Thank you. The paper may now be accepted for publication. 

Reviewer 3 Report

Authors provided all changes needed.